# Inverting the deficit model in global mental health: An examination of strengths and assets of community mental health care in Ghana, India, Occupied Palestinian territories, and South Africa

**Kaaren Mathias**[1], **Noah Bunkley**[2], **Pooja Pillai**[3], **Kenneth A. Ae-Ngibise**[4], **Lily Kpobi**[5], **Dan Taylor**[6], **Kaustubh Joag**[7], **Meenal Rawat**[3,8], **Weeam Hammoudeh**[9], **Suzan Mitwalli**[9], **Ashraf Kagee**[10], **Andre van Rensburg**[11], **Dörte Bemme**[12], **Rochelle A. Burgess**[13]*, **Sumeet Jain**[8], **Hanna Kienzler**[14], **Ursula M. Read**[15]

**1** Faculty of Health, University of Canterbury New Zealand, Christchurch New Zealand and Burans, Herbertpur Christian Hospital, Emmanuel Hospital Association, New Delhi, India, **2** Faculty of Medical and Health Sciences, University of Auckland, Auckland, New Zealand, **3** Burans, Herbertpur Christian Hospital, Emmanuel Hospital Association, New Delhi, India, **4** Kintampo Health Research Centre, Kintampo, Ghana, **5** Regional Institute for Population Studies, University of Ghana, Accra, Ghana, **6** Executive Secretary, Mind Freedom, Accra, Ghana, **7** Centre for Mental Health Law and Policy, Indian Law Society, Pune, India, **8** School of Political and Social Science, University of Edinburgh, Edinburgh, Scotland, **9** Institute of Community and Public Health, Birzeit University, West Bank, Occupied Palestinian Territories, **10** Stellenbosch University, Stellenbosch, South Africa, **11** Centre for Rural Health, University of Kwazulu-Natal, Durban, South Africa, **12** Centre of Society and Mental health, King's College London, London, United Kingdom, **13** Institute for Global Health, University College London, London, United Kingdom, **14** Department of Global Health and Social Medicine and Centre for Society and Mental Health, King's College London, London, United Kingdom, **15** Warwick Medical School, University of Warwick, Warwick, United Kingdom

* r.burgess@ucl.ac.uk

**Data Availability Statement:** The country profiles that were used as a basis for this paper are

## Abstract

Global mental health [GMH] scholarship and practice has typically focused on the unmet needs and barriers to mental health in communities, developing biomedical and psychosocial interventions for integration into formal health care platforms in response. In this article, we analyse four diverse settings to disrupt the emphases on health system weaknesses, treatment gaps and barriers which can perpetuate harmful hierarchies and colonial and medical assumptions, or a 'deficit model'. We draw on the experiential knowledge of community mental health practitioners and researchers working in Ghana, India, the Occupied Palestinian Territory and South Africa to describe key assets existing in 'informal' community mental health care systems and how these are shaped by socio-political contexts. These qualitative case studies emerged from an online mutual learning process convened between 39 academic and community-based collaborators working in 24 countries who interrogated key tenets to inform a social paradigm for global mental health. Bringing together diverse expertise gained from professional practice and research, our sub-group explored the role of *Community Mental Health Systems* in GMH through comparative country case studies describing the features of community care beyond the health and social

available on the data repository Figshare with this link: 10.6084/m9.figshare.24313768.

**Funding:** This work was supported by Faculty of Health, University of Canterbury funding to KM. The funders had no role in study design, data collection and analysis, decision to publish, or preparation of the manuscript.

**Competing interests:** The authors have declared that no competing interests exist.

care system. We found that the socio-political health determinants of global economic structures in all four countries exert significant influence on local community health systems. We identified that key assets across sites included: family and community care, and support from non-profit organisations and religious and faith-based organisations. Strengthening community assets may promote reciprocal relationships between the formal and informal sectors, providing resources for support and training for communities while communities collaborate in the design and delivery of interventions rooted in localised expertise. This paper highlights the value of informal care, the unique social structures of each local context, and resources within local communities as key existing assets for mental health.

## Introduction

Global Mental Health [GMH] organises to address access to psychiatric treatment and human rights violations as central concerns in low- and middle-income countries [LMIC] [1]. Solutions have largely prioritised closing the 'treatment gap' by drawing on technical expertise to scale up 'evidence-based' interventions [1,2]. Mostly, such interventions are rooted in deficit-based approaches to mental health treatment in LMIC [the 'gap'] and may sideline existing community resources that support mental health and care.

The deficit framing is also seen where communities are instrumentalised, ignoring the active and dynamic role they can play in shaping processes and outcomes for mental health [3,4]. Furthermore, an interest in local scale-up to date has not fully engaged with the wider socio-political landscapes within which everyday life is negotiated for people living with mental illness and their families [5].

In recent years, calls to attend to the social determinants of mental health have become more prominent in global mental health debates, as reflected in the Lancet Commission on global mental health and sustainable development [6–8]. This contributed to the 'reframing' of global mental health from a focus on the 'treatment gap' to mental health as a 'global public good' in line with the Sustainable Development Goals. In addition, a growing emphasis on decolonisation, pluralistic interventions, and an "all of society approach" to mental health care within the field of global mental health invites fundamental changes in the ways in which the problems and solutions of mental health may be conceptualised and actioned [9]. Community health systems is a term developed by Schneider and Lehman, who define them as "the set of local actors, relationships, and processes engaged in producing, advocating for, and supporting health in communities and households outside of, but existing in relationship to, formal health structures"[10]. Therefore, in this study, a community mental health system is understood as an extension of the usual way 'system' is used in health care provision, to include both vertical [hierarchical] and horizontal [relational] elements like networking, trust and reciprocity. These community systems cross the boundaries of public agencies, levels of government, and the public, private, non-profit and civic spheres and are by definition "context specific and influenced by local histories, economic and political systems, and social–cultural norms" [10].

What might be possible if we resist the deficit framing that permeates mainstream approaches to mental health and development [11]? In this paper, we suggest that through centring the survival tactics and strengths of people with lived experience of mental illness, families and others providing care and wider communities of groups and individuals, we can develop mental health care that is more relevant, acceptable, sustainable and equitable [7,12,13]. Here, we are inspired by critical Southern scholars who have long articulated the

importance of centring embodied knowledges and knowledges born through struggle which point us to the necessary work we must do in understanding and responding to power structures as the core of efforts for change [14–17].

In this article, we analyse four diverse settings to disrupt the deficit model in GMH, such as in the WHO Mental Health Atlas, which emphasises the lack of biomedical and psychosocial care by unfavourable comparisons between health system resources in high- and low-income settings [18]. This approach, which typically holds high-income countries in the global north as the gold standard, can perpetuate harmful hierarchies and make assumptions [7] Instead, we highlight assets for community mental health within different complex local realities, and seek to advance scholarship and policy through a strengths-based approach to re-imagining community mental health care. Rather than seeing 'community mental health' as a rational, cost-effective 'system' of formalised 'services', we make visible community mental health as emergent within situated social practices and relationships [5]. To examine community mental health practices in Ghana, India, the Occupied Palestinian Territory [oPt] and South Africa, we draw on the experiential knowledge of community mental health practitioners, advocates and researchers living, working and conducting long-term relational research in these locations. We posed two questions:

- How does the socio-political context influence the practices and processes of community mental health systems in four different locations?

- What common assets, including contextually specific practices of 'informal' mental health care exist in these communities?

## Background

Mental distress and mental health care design are influenced by often partially acknowledged political, social and economic forces that reflect a larger relationship between biomedicine, governments and technocrats responsible for social security and economic growth [7,19,20]. These socio-political histories are embedded into communities and shape mental health systems, community assets and the lived experience of mental health and illness[20,21]. Communities are the product of particular historical and economic conjunctures, which in many low-middle-income countries [LMICs] contain the traces of colonial impositions of governance, institutions and boundaries [22,23].

Within GMH, solutions proposed in the first decade of the 21$^{st}$ century aimed to intervene in places 'where there is no psychiatrist' [24]. One approach was by 'task-shifting' to community health workers and integrating mental health into primary healthcare [25]. Another was the development of streamlined, 'evidence-based' 'packages of care' that included psychiatric and psychosocial interventions that could be tested in community trials and rendered scalable across sites, countries, and regions [26–28]. With this focus on formalised, replicable interventions developed through standardised training in technical knowledge and skills, the contribution of so-called 'informal' care delivered by families, neighbours, healers and religious leaders has often been poorly acknowledged, integrated or supported by formal health and social care systems[29–31]. As a result, the division between 'formal' and 'informal' systems creates a hierarchy in which formal health and social care is defined as that delivered by -trained professionals backed by relatively strong governmental and, in LMICs, donor support. Informal sources of care and support, provided by what are considered to be 'untrained' or 'lay' persons with limited expertise, are seldom recognised as having equally valuable skills and knowledge in their own right [30]. Table 1 provides an overview of aspects of the formal mental health services provided in our four case study countries. Notably, numeric comparisons of mental

**Table 1. Profiles of Ghana, India, Occupied Palestinian Territory and South Africa, comparing existing biomedical services [12,31,33–41].**

| Item/Country | Ghana | India | Occupied Palestinian Territory | South Africa |
|---|---|---|---|---|
| Population [in millions] | 30,832 | 1,390 | 5.39 | 60.04 |
| Number of community health workers per 10k | 6.12 | 6.94 | - | 9.16 |
| Number of psychiatrists per 100,000 population | 0.07* | 0.75* | 32 | 1.52* |
| % of Health budget allocated to Mental Health | 1.4 | 0.8 | 2.5 | 5 |
| Status of MH policy and / or legislation | Mental Health Act 846 2012 Mental Health Policy 2019–2030 Mental Health Care Strategic Plan 2019–2022 | Mental Health Care Act 2017 | None | Existing National Mental Health Policy Framework and Strategic Plan 2013–2020 |

health service provision in global mental health typically highlight formal care as the measure of value, such as the number of trained community health workers or psychiatrists. In these comparative metrics, informal supports are less measurable and more fluid, and thus their potentially transformative contributions are rendered invisible and underacknowledged. However, it is worth noting that in some respects this informal/formal divide is increasingly blurred. Under neoliberal market logics of diversified service delivery and diminished state-funded provision, NGOs, framed as civil society and 'grassroots' providers, are increasingly integrated into formal systems, and receive significant donor backing and funding as described below. In addition, with limited investment, 'formal' health services rely on 'informal' practices, such as philanthropic donations of essential medicines and other supplies [32].

While some have argued that transitions to community-based care, particularly in the context of the treatment of people with serious mental illness in the global south, is informed by a colonial logic designed to reduce dependence on state investment this overlooks the ways in which community-led care could extend beyond economic and medicalised rationales focused on increasing access to biomedical treatment to a more expansive logic of care that includes addressing social, political and structural sources of violence and exclusion [22,42]. Ultimately, community led-care could lead not simply towards the delivery of community-based treatment, but a re-articulation of what counts as care in the first place.

By focusing on a system's inadequacies, the deficit framing of Global Mental Health [GMH] overlooks the influence of local socio-political and cultural contexts or fails to create space for interventions that build on community strengths and assets [5]. Many critical scholars have called for nuanced approaches that attend to structural, cultural and social factors influencing mental health conditions, care and recovery and that engage with lived experiences within the specific contexts of local communities [12,43–47].

As GMH policy pushes for a renewed shift from specialised institutions to community-based services, a closer context-based conceptualisation of the role of communities in mental health care is central. Communities are often defined by physical geography, constituting the boundaries of the space within which services are delivered, as in the concept of 'going to the community' [19,48]. While they are assumed to share values, in practice, communities of place may include considerable diversity, presenting challenges to simplistic harvesting of community resources for mental health care. Furthermore, drawing on more diverse forms of community, which cohere around relationships, and practices, may offer a more holistic approach to mental health [49]. In the remainder of this paper, we focus firstly on examples of the historical, social and political influences on community mental health in our four selected countries

before describing assets and informal care for community mental health in each context. In our analysis, we highlight what is distinct to each setting, as well as shared aspects linked to historical, political, social and cultural influences.

## Methodology and methods

This work was part of a wider project that aimed to shift the conversation in GMH towards a social paradigm that considers the impact of broader social, cultural, historical and political contexts on mental health. Methodologically, this project also sought to counter epistemic injustice and to work actively across power divides in GMH, such as those between practitioners, academics, activists, and people living with mental health conditions, and between different global locations. Instead of uni-directional knowledge transfers and capacity building, which are often underpinned by hierarchical and colonial assumptions about "whose knowledge counts", the project was grounded in mutual learning. The result was an iterative, slower-paced, and reflexive process that prioritised trust and relationships and remained responsive to specific needs and critiques within the group. Mutual learning enabled us to build sustainable relations, shift our thinking, and develop academic, audio-visual and policy outputs while reflecting on how epistemic power can be better shared in GMH [50]. This is mirrored in our approach to data analysis which used reflexive thematic analysis to acknowledge the subjectivity and positionality of the researchers and employed an interpretative, theoretically informed approach to generating themes [51]. Our detailed process is described in **Box 1**.

### Box 1. Description of the mutual learning platform

The mutual learning process, entitled "Together to Transform", included 39 academic and community-based partners working in 24 countries, with expertise deriving from different locations, academic disciplines, lived experience, research and professional knowledge. We formed smaller thematic "pods" through a collaborative agenda-setting exercise and our group [n = 19] self-selected around the theme of Community Mental Health Systems. Over 14 months, we held 16 online meetings of 90 to 120 minutes each in which we agreed to focus on community assets as important but under-acknowledged facets of health systems and to foreground their histories, contexts and everyday practices of care. We focused on community health systems in Ghana, India, oPt and South Africa where team members had long-term engagement, research and experiential knowledge. While these four countries have diverse populations and geographies, they share aspects of their political economies and colonial histories.

Summaries of community mental health systems in each country were developed using primary data already collected by group members, most of whom had worked together before (see Table A in S1 Text). The affiliations, contributions of co-authors and other supporting team members are summarised in Table B in S1 Text. Collaboratively, the teams defined cross-cutting key parameters of interest, including socio-political, economic and historical factors impacting on infrastructures and lived experiences. To facilitate mapping and reflection, each country 'team' summarised specific assets, formal and informal care, resources and challenges against the backdrop of the historical and socio-political economy using Burgess and colleague's socio-political wheel [Fig 1] which is underpinned by an interest in the intersectionality of varied social locations and experiences [5]. The model is designed to illuminate how social processes and factors that permeate wider societies, intersect to determine the presence or absence of social and

structural determinants of poor health [52,53]. In this way, we shift our attention to how varied social, identity and political parameters [and their intersections] produce lived experiences of exclusion and discrimination, and influence poor mental health outcomes.

Using the dimensions of the wheel as a guide, teams mapped the shared and divergent features of each country by drawing on primary research data [interviews, focus groups, ethnographic fieldnotes] and secondary data that included internal and external reports. The primary research studies all had ethics approvals [see Appendix One. Primary data source studies used for Country profiles. Supplementary sources]. We then drew on our local networks of practitioners and researchers to analyse the four country reports using reflexive thematic analysis [54] to identify and describe the existing informal and formal mental health care services. This approach to analysis pays attention to the subjectivity and positionality of the researcher and the development of themes grounded in shared meanings and concepts and analyzed through theoretical frameworks. Analysis was orchestrated at the country team and larger pod levels by familiarising ourselves with the findings and data, comparing and contrasting findings within countries and across the four countries profiled, and generating key themes from the data.

## Findings

### Contexts of care—the socio-political economy of mental health

In this section, we draw attention to the ways that colonialism and political and economic policy have influenced mental health care. Although critically shaping everyday mental health care, such historical and 'upstream' determinants are seldom detailed.

### Colonialism

British colonial psychiatry responses varied across all four countries but typically promoted a custodial model. Resources were used to build institutions for confinement with 'lunacy laws' placing people whose behaviour was believed to be disruptive or dangerous in prisons [56]. In South Africa, while psychiatric services predominated through urban asylums, this was significantly skewed towards white patients [57]. In India, through the 19th and 20th centuries, government- mental health services were delivered almost exclusively through a handful of large mental health institutions, which extended nationally, first through the East India Company and later through the expansion of Western psychiatry. Associated pharmacological treatments were delivered through inpatient and outpatient services for people with severe mental health conditions, although the total number of psychiatrists and inpatient beds remained low [58]. This focus on institutional services rather than community-based care is still evident in mental health budgets today. In India, 94% of the 2022 direct mental health budget [i.e. funding allocated to the Ministry of Health and Family Welfare] was assigned to just two tertiary health care institutions, leaving 6% for implementing district mental health programmes in the rest of the country [59]. Similarly, in oPt, just 2.5% of the underfunded healthcare budget is dedicated to mental health, of which 73% is directed to the single psychiatric hospital [38]. In Ghana, the country's three dedicated psychiatric hospitals absorb most of the mental health budget and there is no ring-fenced budget for community mental health care.

In South Africa, colonial and apartheid legislation and policies have created and sustained gross inequities between white and non-white population groups, which have led to poor

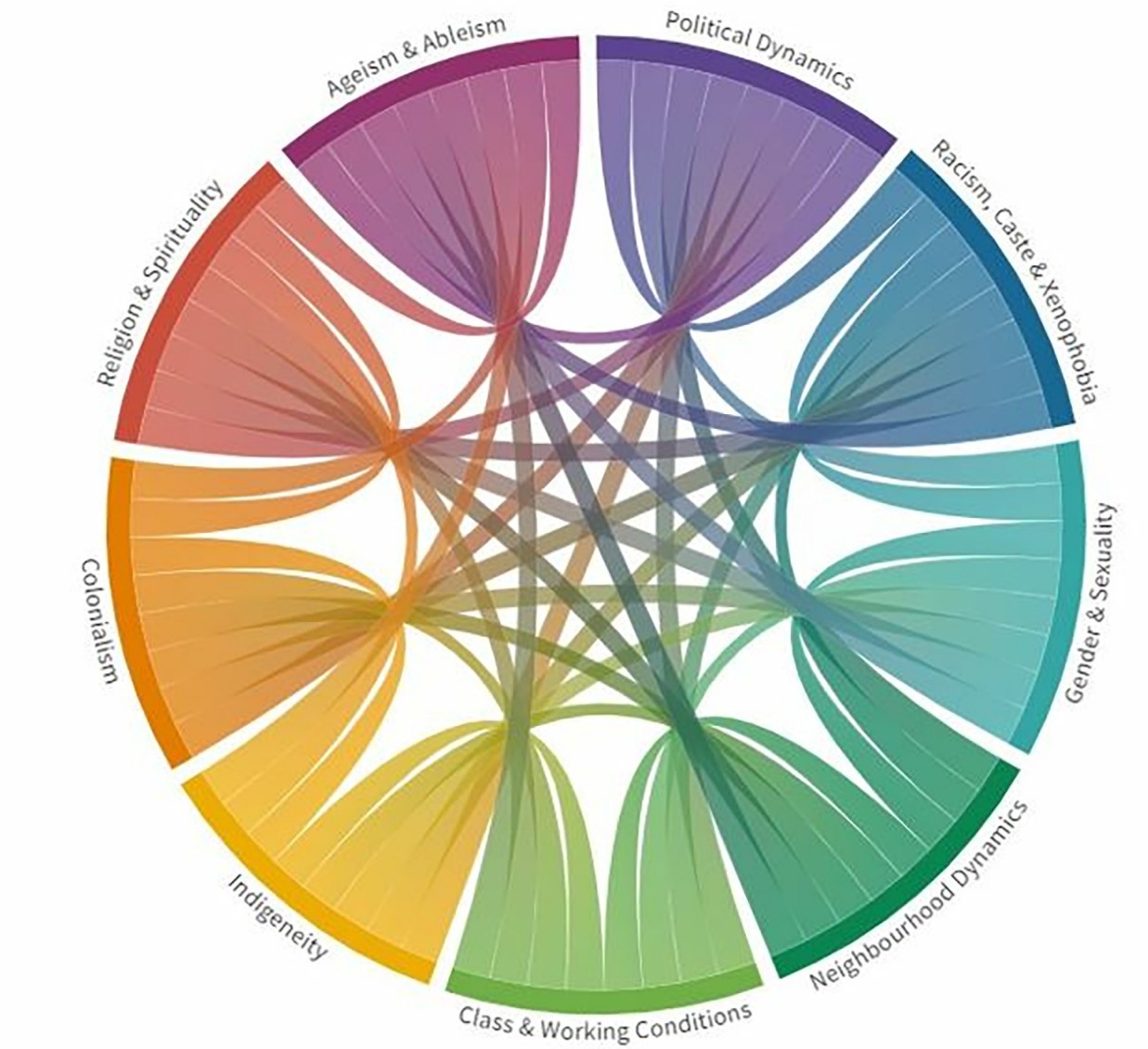

Burgess, R. A., Ní Chobhthaigh, S., Lee, S., Pathare, S., Iemmi, V., & Lund, C. (2022, June 24). Scoping Review Protocol: Establishing a Socio-Political Economy of Global Mental Health. Retrieved from osf.io/e68un

**Fig 1. Socio-political economy of Global Health, Burgess 2023 [55].**

access to and quality of health services, education, socioeconomic opportunity, and capital ownership [60]. Security fencing keeps residential patients contained within institutional boundaries, pursuing notions of danger to the surrounding community [61]. Apartheid also meant that services were significantly skewed to provide for white patients, and mental health services for people of colour, when accessible, were harsh and of poorer quality [62].

Likewise, in Ghana, the underdevelopment of mental health services has roots in these historical legacies and colonial legal frameworks. The 1888 Lunatic Asylum Ordinance governed practices of 'alienation', which confined suspected 'lunatics' initially in prison custody and later in the Accra asylum, which opened in 1906 [56]. It was only after independence that two further psychiatric hospitals were constructed. This left the care of most people with mental

illness to families and communities, even where this meant prolonged use of restraints such as logs or chains [63]. Centuries of the transatlantic slave trade, followed by colonial occupation and, later, structural adjustment policies, established deep and lasting economic and social inequity that feeds into ongoing disinvestment in public health and enduring household precarity. There are chronic funding deficits in the mental health system, resulting in out-of-pocket costs for admission, medication and other treatments. At the same time families have limited resources to meet these costs. For example, an older woman in Kintampo who was caring for her sister, described how her sister's relapse was due to lack of resources, saying *"what resulted in the return of the sickness was that I didn't have money. So because I didn't have money the sickness came again."* [64].

In oPt, the context of settler colonialism and the logic of elimination is central to the context of mental health systems. Palestine has been under Israeli military occupation for over 50 years, with a lack of access to land, water, borders and freedom of movement of people and goods. Furthermore, the protracted Israel-Palestine conflict has exposed the Palestinian population to human rights abuses, including the use of lethal force against civilians, land confiscations, and house demolitions. The current war on the Gaza Strip has, to this date, resulted in over 16,000 people killed, among them 6000 children. Besides carpet bombing the Strip, Israel has placed it under a complete siege, cutting off supplies of water, food, fuel, and medicine. Israeli airstrikes target civilian infrastructure indiscriminately, including homes, roads, water tanks, hospitals and clinics, schools, and places of worship such as mosques and churches. The current onslaught occurs in the context of decades of settler colonialism and its related violence have induced a severe economic crisis responsible for the de-development of public services such as schooling and mental health services, and poverty affecting two-thirds of the population [38,65]. A mental health practitioner explained to team members during an interview:

> *"Of course, the political context [is a challenge], for example sometimes we cannot have access to these [health] Centers, because the Israeli military closed the access. Or sometimes we can have a group activity in one of the Centers and there's tear gas all around."*

## Neoliberal capitalism

Globally, neo-liberal capitalism has brought about a qualitative change in the world economic system that has crossed the boundaries of nation states and undermined local and indigenous health systems [66,67]. Neoliberal capitalism is characterised by commitments to trade liberalisation, privatisation, deregulation, and governance systems that extend competitive markets into all areas of life while showing a fundamental aversion to social collectives and social redistribution. We outline how neoliberal economics has acted as a distal macroeconomic determinant to increase mental distress and reduce the quality and affordability of public mental health care [68,69].

Our country profiles illuminate how changing macroeconomic conditions impact mental health and opportunities for people living with mental illness to enjoy equal rights and opportunities in the community. For example, in Ghana, neoliberal reforms led to economic growth but also higher unemployment and increased inequalities, particularly for young people [70]. The migration of young men from Kintampo and surrounding communities across the Sahara to Libya [and often on to Europe] is one manifestation of this. Unemployment, inequality and associated health determinants such as homelessness and food insecurity have consistently been associated with increased mental health problems like depression and anxiety [71]. The connection between poverty and mental distress is complex and works in both directions due

to social causation and social drift, which act across the life course [once in poverty, you are more at risk for depression, and then you are more likely to remain poor] [72–74]. Other studies have moved beyond association to demonstrate the effect of economic interventions, for example, cash transfer programmes, which have decreased stress and depression [72–74].

The contribution of neoliberal economics to poverty and inequality has become clearer. For example, in a recent analysis of structural adjustment programmes and International Monetary Fund loan conditions across 81 developing countries, income distribution and higher rates of poverty were worsened through mechanisms such as increased costs of basic services, restructured taxation, increased unemployment and reduced government revenue [66,71].

Team members described how neo-liberal economic policies influenced the form and content of public health services through uneven regional development [linked to the economics of transnational corporations] and disinvestment in public services [for example, due to IMF loan arrangements that reduce government spending on public sector wages]. For example, in Ghana, UK development aid has provided funding for the establishment of the Mental Health Authority and the implementation of the Mental Health Act, yet the impact of these programmes is limited by requirements to reduce health expenditure to meet the cost of debt repayments [75]. In addition, health care in countries such as Ghana and South Africa remains dependent on donor funding from high-income countries, but World Bank categorisation impacts the available resources for health care. For example, since Ghana moved to lower-middle income status in 2010, donor contributions to the health budget have more than halved [20,76].

In Palestine, decades of illegal military Israeli occupation, blockade, and apartheid have led to economic devastation, resulting in eroded health and social protection systems [20]. In the Gaza Strip, the current siege and bombardment have brought its health system to its knees, leaving thousands without access to any form of even basic medical care or support [77]. With a lack of medication, electricity, and water supplies, doctors can hardly provide even the bare minimum of healthcare to the population. In both the West Bank and Gaza Strip, there are few resident mental health professionals, making mental health services less accessible.

Neo-liberal economics have also promoted a diversification of health financing and provision, with increasing involvement of the private for-profit sector. Our country profiles described how for-profit health providers have over-investigated, over-diagnosed and over-treated to provide financial benefits to doctors and other providers [78,79]. While for-profit mental health care can be of higher quality in all four case countries, colleagues in North India also described examples of unethical and poor quality private psychiatric care [for example, requesting repeated radiological diagnostic tests], which led to catastrophic expenditure with one father describing how he had to sell his buffalo to pay for his son's brain scan [78]. In Ghana, limited regulation of private health providers, including for drug rehabilitation, could result in dangerous practices such as 'cold turkey' withdrawal, misdiagnosis and unlicensed prescriptions.

The influences of neoliberal capital flows are further evident in how care is provided for people living with serious mental health conditions in communities. In South Africa, in the absence of access to opportunities for income generation and residential care, caregivers and families of people living with severe mental illness often become dependent on a monthly government disability grant. Grant funding support is an essential element of fulfilling the complex needs of people with severe conditions but it can risk becoming part of a system that ranks the investment value of vulnerable populations, determined by global auditing firms. In the almost complete absence of community-based psychosocial support services and almost no prospects to generate income, people with severe mental health conditions risk being rendered as one-dimensional income-generating bodies with little agency within neo-liberal

contexts [61,80]. Instead, a more comprehensive package of psychosocial support services alongside disability payments would better serve people to break the cycle of dependence.

## Responding to the real world—Community assets for mental health

While communities are shaped by global socio-political forces, they also form local, contextualised responses to mental health challenges. Through cross-case analysis and discussions, we identified the following assets that build community resilience and form a bedrock of mental healthcare in these countries. Keeping with our orientation to 'informal' assets beyond health and social care systems, we map these onto the primary social groupings within the study countries: the family and home, community and social support, faith-based care and traditional healers, and non-profit organisations. The influence of historical, social, cultural, economic and political factors is threaded through all these domains, shaping the possibilities for care and support in various ways.

### Family and home

Families and family homes were identified as critical mental health assets in all four countries, providing various forms of practical, social, financial and emotional support. This care is typically taken for granted and unexamined in mental health policy, particularly in LMICs, with limited consideration of how family care is practiced and what it contributes in different contexts. Family care can be actively co-produced. For example, family members in India described how they negotiate and plan activities of daily living with family members with mental health conditions, such as who will water the plants or pick up children from school. The same study described how family care can also be hierarchical, harsh, disregarding and excluding [81]. Household composition and organisation, normative obligations and responsibilities and access to resources vary considerably within and between settings [44]. For example, ethnographic fieldwork with families in Ghana showed how family members with mental health conditions are entitled to shelter and support within the 'family house', a communal living space commonly constructed by a male relative and passed down through generations [64]. The 'family house' can provide a safe space to engage in the daily life of the household and a network of supportive care as all members of the household are involved. Meals are cooked communally, distributing costs among those living there and providing food for those who cannot afford to feed themselves. In Palestine, the family home is the primary communal space where people with mental health conditions can participate in social life while receiving protection, care and support. A non-profit organisation [NGO] worker described:

> "*Families in Palestine are more protective and can deal with somebody with schizophrenia or psychotic symptoms. Families become close and try to help. Maybe the diagnosis is still sometimes secret inside the family–like schizophrenia or psychosis. But they don't hide him from people.*"

All four country profiles also described socio-political factors that influence care by families. For example, in all four settings, care is strongly gendered, with women typically doing the additional work of cleaning, washing, cooking, and feeding family members with mental health conditions. [44,82].

In all four countries, families provided social and financial support. This was leveraged to access informal and formal care and opportunities for participation in income generation and household responsibilities, such as farming, child-minding, livelihood-related tasks and

community participation. For example, one man in Ghana with bipolar disorder who had been dismissed from formal employment gained employment in his mother's bread-making business, enabling him to work in a more flexible and supportive environment [83].

Family members can also play a key role in supporting participation in the social life of the community, including family events such as weddings and funerals, as observed in India and Ghana [74,76]. However, people with mental health conditions can also face prejudice and exclusion within their communities [84]. The extent of involvement in family activities depends on several factors, including the perceived severity of the distress, family attitudes, the person's gender and the nature of the occasion. Care that starts as well-intended can also become overpowering and controlling [84].

Of course, such support is not without cost [85].The lack of government assistance for family caregivers in India and oPt means that caregiving can lead to overwhelming physical, psychological, financial and social obligations. South Africa has a small social support grant for families of people living with mental health conditions, although it can be challenging to furnish the required documentation. In Ghana too, some local government funding is allocated for people with disabilities, and poor households can claim a livelihood support grant. These funds, though small, can prevent destitution. However, the funds can also be difficult to access and cash-strapped or corrupt local governments may divert the disability budget elsewhere. A caregiver in Ghana described how she had attempted to apply for a benefit for poor families without success:

> "*They said they will give to forty people but our names are not part. Even the money they said they were bringing hasn't come yet. [. . .] I've never received anything. I've written [our names] for it two or three times, but they haven't given us anything*" [81,83].

## Community and social support

Beyond the family, wider relationships linked to shared geography or identity play a central role in mental health in the four countries. For example, in rural India and South Africa women bonding over the shared experience of their agricultural work describe reduced stress and greater satisfaction [86,87]. Similarly, within caste groups in India, families can provide high levels of support to each other, for example a husband who had to travel to seek care for his wife described, *"Our neighbours did all the harvesting and caring for the fields while I took my time going with my wife on a pilgrimage to seek mental health care"* [81]. Despite stigma and discrimination within communities, social support networks are important assets that can provide a sense of belonging, meaning and practical support [81].

Formalised 'peer support' in which people with lived experience of mental illness are brought together to provide mutual support and advice is not widely available in many of the study settings [88]. However, our research uncovered organic forms of mutual support among people with lived experience and caregivers. These could cohere within existing social settings, such as caregivers meeting within a prayer camp in Ghana, patients meeting at a mental health clinic or young people living with mental illness encountering others through social media. Collaborators in South Africa and India described informal peer support where a person with a mental health condition would visit another affected person in their community and accompany them to seek care. [81,83,89] Shared experiences of mental health treatment will often encourage greater participation in formal and informal care and foster a deeper sense of purpose and belonging. As a South African service user noted about her neighbour,

> "*What encouraged me is that there were two of us that came to the clinic coincidentally together. Although it wasn't during the same time, but I was encouraged.*"

In Ghana, informal peer support was often nurtured within community spaces such as churches and prayer camps where people stayed together for some time to seek healing. Ethnographic fieldwork revealed how mothers would come together to cook, as well as share advice and encouragement. This organic form of peer support also took place within community mental health clinics where caregivers met when awaiting consultation. In Kintampo, this was formalised into a weekly group meeting, including 'psychoeducation' by community mental health workers. However, it was often the informal, spontaneous sharing of lived experience among caregivers and people living with mental health conditions that was most valued [83].

In oPt and Ghana, there has been increasing involvement of people with mental health conditions in peer support, advocacy and activism [83]. In Ghana, some of this has resulted from interventions driven by international funding. For example, the NGO BasicNeeds began establishing 'self-help' groups as part of its inaugural activities in Ghana in the 2000s [90].More recently, the UK-based 'Time to Change' anti-stigma campaign trained mental health 'champions', which helped to expand and strengthen networks of people living with mental illness and forums for advocacy and peer support have arisen organically through grassroots actions by concerned individuals or groups [91]. For example, following her diagnosis with bipolar disorder in 2016, Abena Korkor set up a WhatsApp group to provide social connection and support for people living with mental health conditions in Ghana. Since then, several predominantly young people with lived experience have made astute use of social media for mental health advocacy and information sharing.

## Faith-based care and traditional healers

Despite different cosmologies and explanatory frameworks for mental illness across all four countries, traditional and faith healers are highly valued sources of advice and treatment for mental illness. In India, people with mental health conditions, as well as practitioners, were often pragmatic and pluralist in their approach and accepted that people go back and forth between traditional healers and biomedical services as described by a doctor [92]:

> "*Well of course devi and devta [local deities] are the most important in this area. . . around 50% of people come first to the community health centre and maybe 50% go first to consult with the gods or mali [traditional healer]*".

Religious organisations also support culturally mediated forms of exclusion for people with mental health conditions in all four countries in this study. For example, in Ghana, India, and South Africa, religious and 'traditional' frameworks commonly attribute mental illness to moral failings, contravention of taboos or manipulation of evil forces such as sorcery, thus reinforcing conservative moral codes. This is often targeted at women who may be blamed for engaging in 'sinful' activity such as adultery or using witchcraft [93]. Traditional and faith healers may invoke exorcism rituals, which can be linked with stigma and discrimination as well as inhumane forms of punishment, constraint and confinement [86,91,94].

However, our team members found that many traditional and faith healers are pragmatic and will often refer people to allopathic care when there has been no improvement. The concept of '*dawa aur dua*' [prayers and medicine] has been used widely by community organisations promoting mental health in India and builds on this pragmatic pluralism [95]. This reflects the pluralist tenets of both Hindu and Muslim healing systems, including Ayurveda, Yoga and Naturopathy, Unani, Siddha, Sowa-Rigpa and Homeopathy [AYUSH], which are a formal Department in the Ministry of Health. In rural North India [95].

In Ghana, a similar 'medicine and prayers' approach is commonplace. Traditional healers of various kinds and faith healers from Christian and Muslim traditions offer a promise of

healing through addressing spiritual problems, and provide valued spiritual counselling and psychosocial support. Although the focus of international agencies has been on human rights abuses, many healers are also taking action to reduce or eliminate restraints, and work alongside families and mental health nurses to offer alternatives [96]. Their ability to address spiritual concerns as well as their availability, accessibility and affordability make them an appealing choice for many [97,98]. Ghana is unique among the countries studied in formalising collaboration with traditional and faith healers within mental health policy [99]. The 2012 Mental Health Act [846] in Ghana promoted a move from an institutional model to community-based mental health services and described a pluralistic approach, including traditional and faith healers, government mental health facilities and NGOs as all contributing to the delivery of mental health care [31]. Notably, however, the government Mental Health Authority retains the privilege of regulation and oversight and is the sole recipient of public funding.

In South Africa, traditional healers are typically more accessible than biomedical providers and often contribute to emotional and spiritual wellbeing [100]. Researchers in South Africa and Ghana described ongoing attempts to better formalise and integrate traditional and faith healers into the broader health system, as a regional coordinator of mental health care in South Africa described [99]:

> "*The services of traditional and faith-based healers are relevant because the majority of our community members still access treatment over there, and we cannot entirely close or prevent them from rendering these services.*"

However, other providers and policy makers expressed concern that seeking care from traditional or faith healers could significantly delay access to allopathic services, negatively impacting the progression of the mental illness.

In Ghana and South Africa, Christian frameworks, rituals and practices, which were introduced by colonial missionaries, now dominate, alongside significant Muslim minorities in Ghana. Faith groups, including Christian, Hindu and Muslim, can be highly valued assets for people with mental health conditions in all four settings and temples, shrines, mosques and churches are places where people gather for prayer, socialising, charity, healing and can form networks for support. Their leadership and congregations can provide advice and comfort, spiritual care and understanding, and physical spaces for shelter, meetings, ritual and retreatthe. As a nurse in South Africa explained: *"the problem that is bothering you, please talk about it to the women leaders in your church. . ."*

Practitioners in Ghana and South Africa also described referrals of people with mental health conditions from nurses to churches, framing them as accessible settings that might also address spiritual needs. In many cases, churches help alleviate the compounding burdens of being female providers to households in the contexts of male-dominated economies. 'Prayer camps' in Ghana create a space where people with mental health conditions and family caregivers can stay for a period to seek shelter, spiritual guidance, social support and counselling. Lively Pentecostal church services provide a valued social and emotional outlet, particularly for poor or marginalised women whose daily lives can often offer little relief. Prayer offers hope for healing and connects with the intentions of others in the congregation who join together to seek healing and comfort. A mental health provider in oPt explained that places of worship were often more accepting of people with mental health conditions, saying, *"And this is something which is very positive here. I feel there is a sense of accepting also the other differences, you know. And there's also the sense, [. . .] that we have to take care of others who are in need."*

Religious groups can also provide important access to resources such as charitable donations from the leadership or congregations or connections for work. For example, a very poor

woman in Ghana living with a long-standing psychotic disorder was cared for by her elderly mother. With very few social connections and unable to generate income due to stigma and her relapsing illness, she eventually took shelter in several churches. A 'church sister' ultimately offered her employment, and alongside medication from a community clinic, this has enabled her to remain well.

## Non-profit and civil society organisations

A wide range of non-profit NGOs and civil society organisations operate in all four countries. In the study countries, mental health NGOs have become increasingly prominent, reflecting the neo-liberal orientation to a 'mixed economy' of health care delivery and the 'development' model of mental health care. Despite the common framing of NGOs as grassroots representatives of 'civil society', larger, more established mental health NGOs operate increasingly in ways that blur the line between 'formal' and 'informal' care. They often carry a substantial service burden in the absence of government systems, providing services that include residential care, medical services, protected labour, social network strengthening, and advocacy [101]. Many NGOs are also integral as implementation partners for global mental initiatives [95]. In oPt, NGOs primarily conduct family therapy or community-based psychosocial support among marginalised communities and coordinate with other community resources such as schools and youth clubs. NGOs are typically trusted actors that refer to specialised government services and are involved in awareness-raising campaigns to reduce stigma.

However, NGO services in all settings are unevenly dispersed, meaning services are often inequitably distributed. For example, many metropolitan areas of India have active NGOs providing mental health care, but very few work in rural areas and there has been little coordination across NGOs regionally or nationally for advocacy although this is changing with the engagement of some larger philanthropic funders such as Mariwala Health Initiative [102]. In South Africa, while 2,000 mental health NGOs receive government subsidies, there is poor coordination between government departments and NGOs, meaning their care is fragmented. In contrast, Ghana has relatively few NGOs focused on mental health, with BasicNeeds Ghana the largest nationally. Still, they are steadily increasing in number, with people with mental health conditions, health professionals and religious leaders forming new NGOs or advocacy groups. This creates a tension between the need for equitable access to the forms of support typically provided by such NGOs, such as livelihood support, self-help groups or psychological interventions, and the need for flexible and creative responses in particular communities. As NGOs expand and are increasingly integrated into formal healthcare systems they may have more access to resources and thus be able to extend their operations. However, there is also a risk they become less attuned to the needs of people on the ground and more accountable to the priorities of their funders which may compromise their independence as representatives of civil society.

Beyond these larger and more visible NGOs, across all four countries, there are many smaller charitable groups and individual philanthropists who draw primarily on local donations and fundraising activities and the contributions of local volunteers and supporters, rather than international donors. Faith communities, professional groups, youth organisations, schools and universities, for example, bring together people to offer flexible support to vulnerable people in the community. They engage in many smaller-scale activities to respond to identified needs, such as sponsoring the renovation of mental health facilities, offering clothing, food and shelter to people with mental illness living on the streets, and providing opportunities for employment [102].

In all four countries, these civil society organisations, informal groups and individuals are active mental health advocates, for example, participating in print, radio, TV and social media

to change the narrative around mental illness and providing physical spaces for people to gather and develop advocacy actions to address particular concerns within their communities.

Across all these domains, the informality and flexibility of these supports enable them to respond fluidly and rapidly to people's particular needs. However, that same informality could enable forms of exploitation, abuse and neglect, as there is minimal oversight or accountability. This points to a need to consider how the rights and safety of people living with mental illness could be protected without stifling the innovation, creativity and responsiveness that makes these informal supports so relevant and valued. A recent paper by colleagues in Ghana and Palestine profiled practices for social inclusion and human rights operating within communities and suggested where and how some localised accountability mechanisms operate [84,96].

## Discussion

This slow-moving, collaborative and mutually developed case-study methodology has some significant strengths. The case study method draws on in-depth evidence gathered over significant time periods using different qualitative methods including ethnographic observation, in-depth interviews and repeated data collection at different time points which were analysed, compared and triangulated to provide insights into the specific contexts of the four countries profiled. The diverse representation in the co-author team and others who contributed, provided a breadth of disciplinary, geographic and cultural perspectives [see Table B in S1 Text]. At the same time, challenges and asymmetries in individuals' ability to participate were linked to inequities such as poor internet access and varying institutional recognition and support of this work. The strengths and limitations in the positioning and participation of authors are outlined in greater detail in another paper that emerged from the T2T collaboration [103]. This includes an acknowledgment of the fact that this T2T partnership was initiated by academics located in the Global North reinforcing existing asymmetries in academia and global power relations, that English was the working language of the collaboration which may have excluded some groups and that power hierarchies were active in many aspects of data collection and analysis.

### Considering contexts and assets for co-production with communities

Our findings underscore that programmes and policies need to engage with local historical and socio-political contexts as well as informal care practices to ensure mental healthcare is relevant and acceptable [104,105]. The design and delivery of mental health care in these four countries and elsewhere should be co-designed with communities and build on and strengthen local assets and address locally-identified needs to ensure policies, services and interventions are more acceptable, relevant and equitable [92,104,106]. Co-production of policies or services with people with mental health conditions are not well represented in any of these four community mental health case studies. Co-production involves those with lived experience of mental distress in designing, implementing, delivering and evaluating relevant care, resources and services and ensuring that services are person-centred, cost-efficient, innovative and equitable. It can also benefit people who might use services by valuing their contributions as skilled, capable and experienced actors [106,107].

### Addressing socio-economic and political contexts of mental health

While we have identified NGOs as a potential asset, they risk importing the interests, ideologies and historical colonial power structures of high-income funders, perpetuating colonial and neoliberal mental health systems. With most global health institution headquarters and global health leaders hailing from high-income countries [HICs], there can be a disconnect

between international NGO decision-makers and local priorities [108]. To truly support the co-production of mental healthcare and the strengthening of local assets, there needs to be a shift in global mental health financing so that resources and power are redistributed to ensure mental healthcare systems are run and led by the communities they serve. In addition, sufficient space and support should be given to enable smaller, 'organic' organisations to contribute their learning and experience in equitable ways that respect their community-based experiential knowledge [11].

To address the economic drivers that shape mental health and mental healthcare in these four countries, an equity-in-all-policies approach can ask how policy decisions affect mental health and health systems and those who are structurally disadvantaged. For example, in India policy makers can move beyond addressing farmer suicides by only restricting access to hazardous pesticides, to seeking opportunities to work with the national and international trade sector to address the unfair structures of international tariffs, import restrictions and subsidies in agriculture and their impact on markets for developing country agricultural products and the livelihoods of poor small farmers. In Ghana, a focus on increasing the numbers of mental health workers or criminalising substance use needs to be accompanied by policies to increase access to employment and work opportunities and provide workplace protections and adaptations for those living with psychosocial disabilities [83].

## Redistributing power through asset-based approaches

An asset-based approach values, supports and enables access to finances and can redistribute power to the communities affected. This differs from task-sharing or task-shifting in that it is a holistic approach that is less concerned with integrating or reshaping existing local expertise and resources but maintains a broader lens of how these informal assets are situated within the multiplicity of life as lived within specific communities [109]. This helps to retain the key aspects that make them effective, such as contextual relevance and embeddedness, flexibility and pragmatism, as well as independence from the ponderous wheels of bureaucratic governance. Of course, a lack of governance can also lead to exploitation, corruption and potential abuse, so this flexibility could be joined with localised forms of accountability, for example, through oversight by local citizens' groups or traditional authorities in contexts such as Ghana [110]. Governments and funders must partner with communities and those with lived experience so that community mental health assets can be strengthened and supported and care rendered more relevant to the specific needs of community members. This could lead to new innovations in mental health care, grounded in local assets and a more acceptable and effective way of adapting treatment programs that have worked well in HICs [111]. There is also scope for collaborative care between traditional and faith healers with biomedical providers, an approach that has been trialled with positive mental health outcomes in Ghana and Nigeria [112].

Examples of the former include the Friendship Bench, first initiated in Zimbabwe, which began with recognising the value of support from older women in the community. Termed 'grandmothers', they are trusted community members for people with mental health conditions [113]. More contextually relevant treatment approaches address both the illness itself and the household's ability to refocus and train for improved income-generating competencies, helping families sustain better treatment plans [114]. In Ghana, community mental health workers have joined forces with healers to improve care and support for families and help prevent harmful practices. They have proved astute at mobilising community resources to support their efforts, helping to meet some of the deficits in mental health resources such as medication supplies and countering the effects of stigma by identifying community allies to support social integration and providing work and housing [99].

However, strengthening assets is not a substitute for investments in service improvement or attempts to address the structural causes of health inequities. Enhancing community assets can result in a reciprocal relationship between the formal and informal sectors whereby resources for support and training are provided for communities to strengthen and sustain their informal roles and, in turn, communities inform healthcare interventions with localised expertise [115]. Globally, social prescribing has been suggested as a mechanism to promote this type of dialogue between community systems and health care [116]. However, where these have been implemented so far, the focus has been on individuals and consideration of structural and political dynamics has been non-existent. Recent work in Colombia suggests the positive potential of methodologies that drive community-led action for health improvement using participatory action methods and may be an alternative to the current limitations in social prescribing methodologies [78].

There is also a risk that assets and resources can be instrumentalised in the task-shifting model and lose their unique diversity and relevance by rendering them 'scalable' and measurable. In the process, the very factors which make them successful, as described above, can be weakened or neutralised. For example, the traditional AYUSH medicine systems in India have been used by Hindu nationalist government to give asymmetrical attention to Hindu over Islamic providers [117]. While it is necessary to acknowledge these assets as foundational elements of health systems, it is essential for them to retain their independence and situational relevance so that they are acknowledged but not 'interfered with'. Formalisation imposes a hierarchy where informal assets become co-opted by governments, losing their ability to remain impartial and advocate externally to the healthcare system, as has happened in some approaches to the concept of mental health 'recovery', where a health system can 'bureaucratise' a concept so that it loses the original meaning and intent [118]. Informality is a strength that allows local assets to adapt to changing demands and challenge formal power structures.

## Conclusions

In this paper we have argued for an inversion of the progressive GMH narrative to identify and challenge the sociopolitical structural drivers of mental illness and mental health care system weaknesses while recognising and strengthening community assets that support mental health. Here, we have identified colonisation and capitalism as two prominent examples of the structural political-social determinants actively shaping community mental health care systems in Ghana, India, Palestine and South Africa. Similarly, we have identified the situated practices and resources that can strengthen community mental health care and provide support, including those available within families, communities, traditional and faith-based healers and NGOs. While we have focused on the countries of Ghana, India, Occupied Palestinian Territory and South Africa, to highlight the importance of identifying how such informal care is practised in specific contexts, our findings suggest the need to recognise the value and potential of assets for mental health in communities worldwide.

## Supporting information

**S1 Text. Table A: Primary data sources of studies used for Country Profiles.** Table B: Contributors Table.
(DOCX)

## Author Contributions

**Conceptualization:** Kaaren Mathias, Kenneth A. Ae-Ngibise, Lily Kpobi, Dan Taylor, Kaustubh Joag, Meenal Rawat, Andre van Rensburg, Dörte Bemme, Rochelle A. Burgess, Sumeet Jain, Hanna Kienzler, Ursula M. Read.

**Data curation:** Kaaren Mathias, Noah Bunkley, Pooja Pillai, Kenneth A. Ae-Ngibise, Dan Taylor, Kaustubh Joag, Meenal Rawat, Hanna Kienzler, Ursula M. Read.

**Formal analysis:** Kaaren Mathias, Noah Bunkley, Kenneth A. Ae-Ngibise, Kaustubh Joag, Weeam Hammoudeh, Rochelle A. Burgess, Hanna Kienzler.

**Funding acquisition:** Kaaren Mathias.

**Investigation:** Lily Kpobi, Weeam Hammoudeh, Suzan Mitwalli, Ashraf Kagee, Andre van Rensburg.

**Methodology:** Kaaren Mathias, Pooja Pillai, Kenneth A. Ae-Ngibise, Andre van Rensburg, Dörte Bemme, Sumeet Jain, Hanna Kienzler.

**Project administration:** Kaaren Mathias.

**Writing – original draft:** Kaaren Mathias, Noah Bunkley, Pooja Pillai, Dörte Bemme, Rochelle A. Burgess, Ursula M. Read.

**Writing – review & editing:** Kaaren Mathias, Noah Bunkley, Pooja Pillai, Kenneth A. Ae-Ngibise, Lily Kpobi, Dan Taylor, Kaustubh Joag, Meenal Rawat, Weeam Hammoudeh, Suzan Mitwalli, Ashraf Kagee, Andre van Rensburg, Dörte Bemme, Rochelle A. Burgess, Sumeet Jain, Hanna Kienzler, Ursula M. Read.

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
