## [Decision Letter · Decision Letter 0]

25 Aug 2023

PGPH-D-23-00993

Inverting the deficit model in global mental health: An examination of strengths and assets of community mental health care in Ghana, India, Occupied Palestinian territories, and South Africa.

Dear Dr. Mathias,

Thank you for submitting your manuscript to PLOS Global Public Health. After careful consideration, we feel that it has merit but does not fully meet PLOS Global Public Health’s publication criteria as it currently stands. Therefore, we invite you to submit a revised version of the manuscript that addresses the points raised during the review process.

We look forward to receiving your revised manuscript.

Kind regards,

Cristian R Montenegro

Academic Editor

Journal Requirements:

1. Please amend your online detailed Financial Disclosure statement. This is published with the article. It must therefore be completed in full sentences and contain the exact wording you wish to be published.

a) State the initials, alongside each funding source, of each author to receive each grant. For example: "This work was supported by the National Institutes of Health (####### to AM; ###### to CJ) and the National Science Foundation (###### to AM)."

2. Please ensure that the funders and grant numbers match between the Financial Disclosure field and the Funding Information tab in your submission form. Note that the funders must be provided in the same order in both places as well.

3. Please update your online Competing Interests statement. If you have no competing interests to declare, please state: “The authors have declared that no competing interests exist.”

4. In the online submission form, you indicated that "The country profiles that were used as a basis for this paper are available from the corresponding author on reasonable request.". All PLOS journals now require all data underlying the findings described in their manuscript to be freely available to other researchers, either 1. In a public repository, 2. Within the manuscript itself, or 3. Uploaded as supplementary information.

Additional Editor Comments (if provided):

Dear Authors,

I write to you as the main editor for your submission entitled "Inverting the Deficit Model in Global Mental Health: An Examination of Strengths and Assets of Community Mental Health Care in Ghana, India, Occupied Palestinian territories, and South Africa."

First and foremost, I appreciate your ability to add contextual nuance to a notion that remains as vital as ever to the global development of adequate mental health systems. The notion of community has been central in mental health discourse. Yet, your approach to reconstructing it, based on the vital local dynamics of four distinct countries, is bold and necessary.

The framework of mutuality that you have utilised to develop the piece not only gives weight to the local contexts but also enriches the global understanding of community mental health care. Your work sets a standard of situated, co-produced work in global mental health that should shift the field. As an academic editor, it is a pleasure to be involved in the review of such a seminal piece.

That said, we have received insightful comments from three reviewers who have engaged with your work in depth. While they appreciate the quality and the unique perspective of the manuscript, they have also raised several important points that require your attention, especially reviewers 2 and 3.

Please address the points raised by reviewers either by incorporating necessary changes or by providing reasonable justifications for maintaining your current stance. Please find the detailed comments from reviewers attached for your consideration.

Thank you for your submission, and I look forward to your revisions.

With warm regards,

Cristian Montenegro.

Academic Editor

Reviewers' comments:

Reviewer's Responses to Questions

**Comments to the Author**

1. Does this manuscript meet PLOS Global Public Health’s publication criteria? Is the manuscript technically sound, and do the data support the conclusions? The manuscript must describe methodologically and ethically rigorous research with conclusions that are appropriately drawn based on the data presented.

Reviewer #1: Yes

Reviewer #2: Yes

Reviewer #3: Yes

2. Has the statistical analysis been performed appropriately and rigorously?

Reviewer #1: N/A

Reviewer #2: N/A

Reviewer #3: N/A

3. Have the authors made all data underlying the findings in their manuscript fully available (please refer to the Data Availability Statement at the start of the manuscript PDF file)?

Reviewer #1: No

Reviewer #2: Yes

Reviewer #3: Yes

4. Is the manuscript presented in an intelligible fashion and written in standard English?

Reviewer #1: Yes

Reviewer #2: Yes

Reviewer #3: Yes

5. Review Comments to the Author

Reviewer #1: Dear Author, This paper is well written with the fashion and realistic perspectives however would like to get some clarification.

1. How does the team came to conclusion of selecting 4 country case studies where 24 countries researchers are participate in the T2T programme.

2. How did the research team ensure that team members are not biased when writing interpretive case studies or sharing site information for case studies from each site.

3. Is there possibility to provide basic information on field or community researchers who collected primary data through in-depth interviews and Focus group discussion.

4. Is there reason to leave out the influence of socio-cultural practice in informal care.

Suggestion:

Write on limitations of these case studies

Improve the methodology by providing details on process note on this collaborative case studies.

Describe how did these themes arrived from the data or discussion

Discuss on political aspects in informal care (How local governance can help)

Reviewer #2: This paper starts from the position that the recent history of global mental health has tended to focus on the deficits in available formal health service provision, without acknowledging and valuing the important contribution of informal support that is provided to people with mental health needs in communities. It outlines the dominance of research and measurement of available support in the formal health/psychiatric sector; a point which is well justified, but there should perhaps be some recognition of the recent focus on social determinants of mental health, and the widespread adoption in global mental health research and in WHO and other UN guidance, of lived experience participation (though these principles are far from widely enacted in service provision in countries). This has started to provide some counterbalance to the colonial histories that are well, if sometimes frustratingly briefly, described for such important issues. For example it is incomplete to leave the history of asylums in India with the East India company (even in a brief overview).

While still very surprising, and true to the point being made, the statistic that 94% if the Indian mental health budget is allocated to just 2 institutions is not correct (line 194). This refers to the percentage of the direct budget only.

The use of the term 'care' sometimes grates when referring to abusive systems (lines 204/205 are examples).

The arguments about the negative impact of structural adjustment, capitalism and neo-liberal economics provide a fair broad framework for understanding the context for some of the background drivers of mental health issues in countries, but the mechanisms of links between poverty, unemployment and economic inequity are not described in detail. Making these links explicit is important, otherwise the paper risks moving into polemic, especially as the range of examples of failure of the mental health system lacked a sense of direction. Is the key point the lack of investment, that the type of investment has been harmful, or that injustices of colonial history, neo-liberal capitalism or occupation are drivers of mental ill health, even as (not enough) international investment is made in service provision. Some points are reasonable in themselves, but do not provide a rounded view of an issue, but rather are used to make a single isolated point. For example, the reduction of welfare benefits to being a system whereby people are viewed as 'one-dimensional income generating bodies' does not do justice to the evidence of benefits of social protection for people with enduring and disabling mental ill health - something that has been hard fought for.

There is a typo on line 273 making the sentence unclear.

The paper is strongest in the section describing the influence of non-formal sectors. The case studies add enormously to this, and it would be important that they are easily accessible to the reader. The organising into families, communities, faith-based provides and NGOs feels logical, though it is not clear what the method of analysis was to reach this framework. Peer support is mentioned positively throughout these results, and given the current interest in this area in research, it was slightly surprising that this was not identified as a separate section to report on. A clearer explanation of how the frame for reporting results was arrived at would help the reader to understand questions like this. The Country Profiles are very strong, and more direct comparisons of similarities and differences would have been a useful way to anchor the findings more concretely in the data.

While it is not the primary purpose of this paper to assess quality of care in different settings formally, given the premise of the paper (that community settings provide care that forms an under-appreciated asset), some comment on the degree of human rights protection or abuse in different settings (including access to the right to health) would have been useful. This does arise in the discussion where oversight of interventions is suggested by the informal sector through localised accountability mechanisms.

In the discussion, the point that any development of projects (or research work) should properly understand the community assets in the context of the new provision is well made, and directly follows on from the results. The assertion of the importance of co-production in the discussion is important, though not so clearly coming out of the analysis results. Perhaps the logic of this link could be made more strongly.

The discussion is strong when it attempts to link the use of a community asset approach with engaging with formal services, recognising that these formal and informal assets are not opposed to each other or mutually exclusive. Evidence shows that people make active choices about where they seek support, and often manage to combine accessing care from different sectors. Reference to the recent Gureje et al trial of primary care and traditional healer collaboration would have made sense.

The conclusion is clear and makes the case for inverting the current model of GMH well, though recognising and celebrating the substantial gains already made in the field (largely driven but critical literature and representative oganisations of sevice users) would also be fair and aligns to the discussion. The principle of understanding and mobilising local assets routinely, while avoiding losing their nuance is an important one, and the assertion in the conclusion that social and structural drivers of mental distress need to be challenged while community assets are strengthened is compelling. These conclusions form a valuable contribution to these live debates.

Reviewer #3: This is an important paper that a) examines the strengths/assets of “informal” community mental health care systems (shaped by sociopolitical context); and b) propose an alternative to conventional GMH interventions: to invert a “deficit model”. The authors rely on findings (experiential knowledge) from Ghana, India, oPt, and South Africa. Core to their argument is the idea that to strength community assets (care from family & community, NGOs, and religious organisations) will support reciprocal relationships between formal and informal sectors, and thus, the co-production of knowledge and practice. Apart from the relevance of the content, the fact that the paper is written as a community of practice is a valuable example of how to challenge some hegemonic academic conventions. I think that starting with the historical and “global” sociopolitical shapers of local community life is a strength of the paper (placing communities as both victims of such violent processes, as well as key agents of resistance to them).

I am a critical scholar based in a geographical region that is different than those examined in the paper. From such positionality, I can offer some feedback on a manuscript that already promises to make a much needed contribution.

1) The paper focuses in four specific countries and in discussions specific to “global mental health”, citing literature constrained to such scope. This makes perfect sense; yet, the authors may want to consider knowledge from other regions (e.g. Latin America, although a Colombian experience is in fact included in the discussion). Especially considering how critical Latin American traditions regarding “community” seem to have have shaped arguments made in some of the cited references; and – I intuit- is epistemologically close to the selected method for producing knowledge in this study. For example, calls to invert “deficit” models have been made many decades ago (e.g. Fals-Borda in sociology; Freire in critical education; Quijano in relation to coloniality; Martín-Baró and Montero, in liberation and community psychology; Breilh in critical epidemiology). Of course, very important African and Asian thinkers have made relatively similar arguments, focused on their respective contexts. Overall, diverse authors within community and critical psychology tend to stress the idea of moving beyond deficit models (e.g. Kagat et al., 2019; Berroeta, 2014; Capella & Jadhav, 2020; Greeson et al., 2014, many others). I am not suggesting to cite this work, but to highlight that claims against “deficit” models have been around for a while (and that they have been conveniently ignored by groups with epistemic power, which makes the authors´ argument relevant and necessary when discussing GMH in present times).

2) It may be beyond the main focus of the paper, but claims of themes “emerging” from the data (e.g. p3, p.9) have been contested on epistemological grounds (see Braun & Clarke, 2021).

3) Authors already mention this partially, yet some readers would appreciate a sentence or two explaining more operatively (i.e. at the level of procedure) what exactly is the “sociopolitical wheel” and how was it utilized in the study (p.9).

4) Authors explain well how they produced knowledge; yet, one sentence felt a bit ambiguous: “we drew on our local networks of practitioners and researchers to analyse the four country reports….”(p.9). Which people made up such local networks in each country, specifically?

5) The paper does not seem to include any primary data (i.e. participants´ quotes) from Ghana? (I apologize if it does and I missed it).

6) Again, this point of discussion may go beyond the scope of the paper; yet, the authors discuss colonialism, capitalism and neoliberalism as “determinants” of MH; from the perspective of Collective Health (Breilh, 1977, 2021) – and its articulation into Collective “Mental” Health (Capella, 2023 – in Spanish), these would be part of dialectical processes of “social determination”, not “determinants”.

7) Maybe define more explicitly what is understood as “capitalism” and “neoliberal capitalism”? Many of us know this, and the cited references are appropriate; yet, an explicit operative definition may be useful for some readers.

8) The positive role of NGOs is an important part of findings, as they “carry a substantial service burden in the absence of government systems” (p.22). Authors adequately addressed capitalism and neoliberalism in previous sections; yet, they may want to consider a more explicit link between these and the more problematic aspects of NGOs; e.g. as a well-intended “patch” solution to the inherent contradictions of capitalism (and to specific governments who fail to guarantee basic rights); as potential sources of neocolonial cultural globalization; agents of alienation and ideologisation, etc. (of course, none of this denies their monumental work in helping people where the state fails to do so).

I enjoyed reading this manuscript, and I thank the authors for the chance to learn from their valuable work.

6. PLOS authors have the option to publish the peer review history of their article (what does this mean?). If published, this will include your full peer review and any attached files.

**Do you want your identity to be public for this peer review?** For information about this choice, including consent withdrawal, please see our Privacy Policy.

Reviewer #1: No

Reviewer #2: **Yes: **Julian Eaton

Reviewer #3: No

---

## [Editor Report · Decision Letter 1]

3 Nov 2023

PGPH-D-23-00993R1

Inverting the deficit model in global mental health: An examination of strengths and assets of community mental health care in Ghana, India, Occupied Palestinian territories, and South Africa.

Dear Dr. Burgess,

Thank you for submitting your manuscript to PLOS Global Public Health. After careful consideration, we feel that it has merit but does not fully meet PLOS Global Public Health’s publication criteria as it currently stands. Therefore, we invite you to submit a revised version of the manuscript that addresses the points raised during the review process.

We look forward to receiving your revised manuscript.

Kind regards,

Cristian R Montenegro

Academic Editor

Journal Requirements:

2. Please provide separate figure files in .tif or .eps format only and remove any figures embedded in your manuscript file. Please also ensure all files are under our size limit of 10MB.

Additional Editor Comments (if provided):

Dear Authors,

I hope this message finds you well. I am writing to you in my capacity as editor to discuss the revised version of your manuscript, which has been re-evaluated following your comprehensive response to the initial peer review comments.

Firstly, I would like to commend you on the considerable effort and thoughtfulness evident in your revisions. The manuscript has substantially improved and effectively addresses and incorporates the reviewers' insights, bringing it closer to the high standard required for publication.

However, there are certain aspects that still require attention to ensure the manuscript's readiness for publication. These areas include both specific textual clarifications and broader conceptual considerations:

Line 112: Add "and" before "make colonial assumptions" to complete the thought.

Lines 152-154: I would reconsider what "invisible" means here. Following from historical work tracing the colonial drive towards community mental health in Africa (Quarshie, N. O. (2022). Psychiatry on a Shoestring: West Africa and the Global Movements of Deinstitutionalization. Bulletin of the History of Medicine, 96(2), 237–265. https://doi.org/10.1353/bhm.2022.0023) and other critiques of the neoliberal and fiscal considerations underlying deinstitutionalisation in the US and other countries, one could argue that the lack of visibility and acknowledgement of informal support is the paradoxical result of the importance of that support. The more a mental health system can rely on spontaneous support, the less inclined to measure and calculate this support. Ideally, that support can be invoked in moral terms, as a set of reciprocal obligations that characterise “community”. This is compounded by the lack of clear criteria to estimate when "mental health provision" is enough.

Line 58: Please clarify the concept of “community mental health systems”. What makes the dynamic support provided by communities a “system”? Is it its integration with a formal network of services? Is it its holistic or integrated nature? This is a unique conceptual proposition of the paper and requires more clarification.

Line 215: Please explain what “textured summaries” are and what makes them different from a “summary”. If this concept has already been operationalised in the literature, please add a reference.

Line 240: Please explain “reflexive thematic analysis”.

Lines 345 – 349 (and others). I wonder if the authors are interested in updating their description of the health system in the oPt, more generally, in response to the current crisis. This is complex due to the ongoing nature of the conflict, so I defer the decision to the authors.

Lines 592 – 606: There are two potentially contradictory arguments in this paragraph. One states that NGOs introduce fragmentation because they are not coordinated with governments. The premise is that a coordinated and unified care system is preferable over a fragmented one. But then there is another argument stating that the integration of NGOs into healthcare systems compromises their independence and voice in the representation of civil society. The premise is that NGOs are valuable because of their independence. There is a tension between these two arguments that should be developed or made explicit.

I also suggest a thorough round of proofreading.

Kind regards

Cristian Montenegro.
---

## [Decision Letter · Decision Letter 2]

9 Jan 2024

Inverting the deficit model in global mental health: An examination of strengths and assets of community mental health care in Ghana, India, Occupied Palestinian territories, and South Africa.

PGPH-D-23-00993R2

Dear Dr. Burgess,

We are pleased to inform you that your manuscript 'Inverting the deficit model in global mental health: An examination of strengths and assets of community mental health care in Ghana, India, Occupied Palestinian territories, and South Africa.' has been provisionally accepted for publication in PLOS Global Public Health.

Best regards,

Khameer Kidia

Academic Editor

Reviewer Comments (if any, and for reference):

Reviewer's Responses to Questions

**Comments to the Author**

1. If the authors have adequately addressed your comments raised in a previous round of review and you feel that this manuscript is now acceptable for publication, you may indicate that here to bypass the “Comments to the Author” section, enter your conflict of interest statement in the “Confidential to Editor” section, and submit your "Accept" recommendation.

Reviewer #2: All comments have been addressed

Reviewer #3: All comments have been addressed

2. Does this manuscript meet PLOS Global Public Health’s publication criteria? Is the manuscript technically sound, and do the data support the conclusions? The manuscript must describe methodologically and ethically rigorous research with conclusions that are appropriately drawn based on the data presented.

Reviewer #2: Yes

Reviewer #3: No

3. Has the statistical analysis been performed appropriately and rigorously?

Reviewer #2: N/A

Reviewer #3: N/A

4. Have the authors made all data underlying the findings in their manuscript fully available (please refer to the Data Availability Statement at the start of the manuscript PDF file)?

Reviewer #2: Yes

Reviewer #3: Yes

5. Is the manuscript presented in an intelligible fashion and written in standard English?

Reviewer #2: Yes

Reviewer #3: Yes

6. Review Comments to the Author

Reviewer #2: Thank you for your detailed work in addressing the points of the reviewers.

Reviewer #3: all comment have been adequately addressed

7. PLOS authors have the option to publish the peer review history of their article (what does this mean?). If published, this will include your full peer review and any attached files.

**Do you want your identity to be public for this peer review?** For information about this choice, including consent withdrawal, please see our Privacy Policy.

Reviewer #2: No

Reviewer #3: No
